# Dietary similarity among jaguars (*Panthera onca*) in a high-density population

**Rebecca J. Foster** ⓘ*, **Bart J. Harmsen**

Panthera, New York, New York, United States of America

* rfoster@panthera.org

## Abstract

Prey remains found in carnivore scats provide generalised dietary profiles of sampled populations. The profile may be biased if individual diets differ and some individuals are over- or under-represented in the sample. Quantifying individual contributions allows us to recognise these potential biases and better interpret generalised profiles. Knowing the dietary differences or similarity between individuals can help us to understand selection pressures and identify drivers of distribution and abundance. Using the results of individual faecal genotyping, we re-interpreted our previously-published generalised dietary profile of an elusive, neotropical felid, the jaguar (*Panthera onca*; Foster et al. (2010)). We quantified individual sample sizes, assessed whether the generalised profile was influenced by the inclusion of scats originating from the same individual and prey carcass (pseudo-replication), and quantified the distribution of prey species among individuals. From an original sample of 322 jaguar scats from a high-density jaguar population in Belize, we identified 206 prey items (individual prey animals) in 176 independent scats representing 32 jaguars (26 males, 3 females, 3 unknown sex). The influence of pseudo-replication in the original dietary profile was minimal. The majority of scats (94%) came from male jaguars. Eight males accounted for two-thirds of the prey items, while 24 jaguars each contributed <5% of the prey items. With few exceptions, the jaguars followed the same broad diet, a 2:1:1 ratio of nine-banded armadillos (*Dasypus noveminctus*), other vertebrates ≤10kg, and ungulates, primarily peccaries (*Tayassu pecari* and *Pecari tajacu*). We noted prey switching between wild and domestic ungulates for individuals spanning protected forests and farmland. This first scat-based study exploring individual variation in jaguar diet highlights the importance of armadillos and peccaries for male jaguars in Belize, the need for research on their roles in supporting high-density jaguar populations, and the need for more data on female diet from across the jaguar range.

## Introduction

Dietary studies help us to understand the ecological needs and flexibility of predator species. This is useful for conserving or managing large carnivores that compete with humans for wild prey or face persecution for attacking domestic species. Historically, the diets of elusive

**Data Availability Statement:** All relevant data are included within the paper and its Supporting Information files.

**Funding:** RJF and BJH were funded through Panthera (www.panthera.org) by the Sitka

Foundation (www.sitkafoundation.org) and the Liz
Claiborne Art Ortenberg Foundation (www.lcaof.
org). The funder provided support in the form of
salaries for authors [RJF and BJH], but did not
have any additional role in the study design, data
collection and analysis, decision to publish, or
preparation of the manuscript. The specific roles of
these authors are articulated in the 'author
contributions' section.

**Competing interests:** The authors have declared
that no competing interests exist.

carnivores have been deduced non-invasively by identifying prey remains found in scats sampled in the field (e.g. tiger, *Panthera tigris*, [1]; grey wolf, *Canis lupus*, [2]; jaguars and pumas, *P. onca* and *Puma concolor*, [3]; leopards, *P. pardus*, [4]; bobcats, *Lynx rufus*, and coyotes, *Canis latrans*, [5]). Assuming that the scats are from a representative sample of individuals, this approach provides a generalised dietary profile of the sampled population. However, because field surveys for scats are generally 'blind', they sample an unknown proportion of the population, representing individuals of unknown demographic or social status, each making an unquantified contribution to the sample. Generalised (or hereafter, 'universal') profiles derived in this way may be biased if significant unaccounted variation exists between individuals or cohorts, and some individuals (or cohorts) are over- or under-represented in the sample [6,7]. Maximising the number of individuals sampled may help; however, depending on the sampling scheme, larger samples may not represent more individuals, rather, repeated samples from the same individuals and even duplicate scats from consumption of the same carcass (pseudo-replication). Recently, the study of carnivore diet has been advanced to the individual level by linking scats or kill sites to specific individuals through the use of individual faecal genotyping or GPS tracking (e.g. [8–10]); or through the use of stable isotope analysis (e.g. [11,12]). This approach can address potential biases in the interpretation of universal dietary profiles (e.g. [6]), and, more broadly, has opened the way for understanding dietary variation within populations, revealing 'generalist' carnivore populations made up of specialists or combinations of generalists and specialists (e.g. leopards, [13]; red foxes, *Vulpes vulpes*, [14]). Individual variation in carnivore foraging and feeding behaviour plays an important role in shaping ecological dynamics, impacting on competition and predator-prey dynamics [15]. Dietary variation may reflect morphological or physiological differences between individuals, or learned behaviours, and may be associated with social status, age, and sex [15]. Knowledge of how carnivore diet varies with age, sex, or social and reproductive status may help managers predict how these groups respond to changes in the prey base, and vice versa (e.g. [10,16,17]). Such relationships might be overlooked if data are limited to universal dietary profiles. In this study, we use individual faecal genotyping to re-interpret our previously-published universal diet of a large and elusive neotropical felid, the jaguar [18], investigate individual dietary variation, and discuss implications for the study, management, and conservation of this near-threatened species.

Across their distributional range, jaguars have been documented to take at least 111 wild species [19]. They are widely considered to be opportunistic predators, displaying greater dietary breadth (increased generalism) with increasing prey richness [20]. However, generalist feeding behaviour at the distributional level may reflect specialisations at the population level (e.g. pumas, [21]), likewise, generalism at the population level may reflect specialisations at the individual level (e.g. leopards, [13]; foxes, [14]). Indeed, in a review of jaguar diet, de Oliveria [22] noted a high degree of geographical variation in prey use, suggesting population-level differences. We know of only one study to date that has investigated individual-level differences in jaguar diet. Using GPS collars to track 10 jaguars to kills sites, Cavalcanti & Gese [9] documented differences between the sexes and between individuals in the proportion of each species killed. While informative, such collaring and tracking studies are invasive and often prohibitively expensive, limiting the number of individuals sampled. Scat analysis studies may offer a simpler approach; however, to date we know of none that have investigated variation in diet between individual jaguars.

Scat-based inferences about carnivore diet depend on (1) a reliable method of identification to distinguish between scats of sympatric species, (2) sufficient scats to adequately reflect dietary richness and breadth, and (3) minimising pseudo-replication within the dataset [6,7,23,24]. For large, highly mobile, elusive carnivores, such as jaguars, often these criteria are

not met, limited by funding, field conditions and the biology of the target species. The cost of implementing high confidence molecular techniques (genetics or bile acid chromatography) to distinguish between carnivore species (or to identify individuals) is often preventatively high, particularly outside of developed countries. The use of low confidence methods such as scat morphology, field sign, and detections from camera traps, while cheaper and easier to implement, have been demonstrated to result in misleading dietary interpretations for sympatric species (e.g. jaguars and pumas, see [23]). For wide-ranging carnivores which often exist at low density, such as jaguars, finding sufficient scats can be challenging (e.g. [25]). This may be exacerbated by difficult search settings such as dense vegetation, and harsh environmental conditions, such as heavy rains and exposure to sun and extreme heat, leading to the rapid degradation of scats; all of which are common features of jaguar habitat. Biases associated with small sample size are further confounded by demographic or social differences in defaecation behaviour and location, resulting in the over-representation of individuals that defecate in conspicuous places (e.g. [26,27]). Because large carnivores take large prey, the same prey individual may be distributed across multiple scats, increasing the likelihood of pseudo-replication within the dataset. This may be minimised by setting criteria for excluding scats which are likely to be non-independent, such as those found close together in time and space and containing the same prey species (e.g. [28,29].

For jaguars, sufficient sample size has been estimated to range from 35 to 50 for detecting the most common prey, and upward of 100 scats to estimate true species richness [24,30–32]. Of 45 scat-based studies of jaguar diet to date, the median sample size is only 35 scats (range 1 to 322 scats); and only three (7%) set criteria for assessing whether the scats could be considered independent (S1 Table). Just over one-third (17/45; 38%) used high confidence methods to assign species (14 genetic, three bile acid, S1 Table). Three (7%) used individual faecal genotyping, but sample sizes were low and individual diets were not presented (9 scats from 7 individuals, 39 scats from ≥12 individuals and 50 scats from 16 individuals; S1 Table). Survey periods for collecting scats spanned up to 12 years, lasting on average 3 years (N = 42 studies with associated data on survey length, S1 Table), increasing the likelihood of sampling multiple individuals or longitudinal changes in factors which may influence diet (prey abundance, aging of the sampled individuals, environmental conditions). Overall, throughout the literature, inferences about jaguar feeding ecology have largely been based on low confidence methods of determining whether the scats were produced by jaguars, small sample sizes relative to the available prey base, and little or no consideration of the relevance or impact of survey length or pseudo-replication within the sample or assessment of how many, or which, individuals contribute to the sample.

Dietary variation within a carnivore population may reflect preferences in prey selection or spatial variation in the availability of prey species, or both [9,33]. Depending on the prey that they hunt and eat, individuals will be exposed to different risks and benefits. There will be trade-offs between energetic gains/losses, encounter rates, and risk of injury, when hunting prey that differ in their aggression, size, rarity, and whether they are wild or domestic. Understanding these dietary differences and trade-offs will help with tackling management and conservation issues. For example, distinguishing whether there are individuals who kill disproportionately more livestock than others, given their encounter rate ('problem individuals',[34]) could help in developing appropriate protocols for predator control. Also, individual-based dietary data will help with assessing the energetic needs of breeding females to evaluate population persistence in the face of overhunting [30,35,36].

The largest sample of jaguar scats analysed to date from a single site comes from the protected forest of the Cockscomb Basin Wildlife Sanctuary, and the neighbouring unprotected lands, in Belize (N = 322, [18]). Within this high-density jaguar population ([35,37]), almost half of the prey items identified in scats were nine-banded armadillos (*Dasypus noveminctus*;

46%). Wild and domestic ungulates (>10 kg) jointly made up one-quarter of prey items (white-lipped and collared peccaries *T. pecari* and *P. tajacu*, 15%; red brocket deer *Mazama americana*, 3%; and cattle and sheep, *Bos Taurus* and *Ovis aries*, 8%). The remaining quarter comprised species ≤10kg, primarily the small (2–5 kg) group-living, white-nosed coatis (*Nasua narica*, 11%), and 13 additional taxa (17%) each contributing <5% of the prey items. Based on these findings, Foster et al. [18] speculated that jaguars opportunistically exploit abundant armadillos in the study area, supplementing their diet with large ungulates in order to fulfil their energetic needs. However, in the absence of individual-based data we do not know whether the majority of sampled jaguars ate armadillos and large ungulates, or whether a few armadillo and/or peccary specialists contributed disproportionately to the dataset. Individual-level genotyping of this large dataset offers a unique opportunity to explore the distribution of these prey species among individuals. Knowing that armadillos made up almost half of the universal dietary profile, with the remainder split between ungulates and small vertebrates, we investigated how these three groups were distributed among individuals. Additionally, we explored the effect of minimising pseudo-replication present in the original dataset by identifying and excluding non-independent scats and comparing the universal dietary profile between the two datasets. As long-term camera-trap data from the study site revealed male-biased use of the trail system [37], we anticipated a male bias in the scat data. To our knowledge, this is the first scat-based study to explore individual variation in jaguar diet.

## Study area

The study area spanned the eastern half of the protected lowland subtropical broadleaf forest of the Cockscomb Basin, and the mosaic of unprotected habitats and land-use systems extending from the eastern border of the forest towards the Caribbean Sea, across ~525 km$^2$ (see [18]). Historically, the basin was logged, creating dense secondary forest, and received protected status in 1986. Today, the Cockscomb Basin Wildlife Sanctuary encloses 425 km$^2$ of forest, which supports high-density populations of jaguars and pumas [37,38]. The protected forest is partially buffered from human development by a band of unprotected forest, together forming a contiguous forest block. The unprotected fragmented landscape to the east of this forest block comprises a patchwork of pine savannah, shrub land and forest, interspersed with villages, milpa ('slash and burn') farms, fruit plantations, cattle pastures and a single highway running north-south [18]. The region supports a diversity of potential prey, including at least 23 species of wild terrestrial mammals (>2kg), in addition to domestic animals such as cattle, sheep, pigs and dogs outside the protected lands. Although the Cockscomb Basin Wildlife Sanctuary is protected, the periphery is subject to illegal incursions by game hunters, and unregulated hunting is commonplace throughout the unprotected landscape.

## Methods

A previous investigation into the feeding habits of jaguars and pumas collected 645 carnivore scats opportunistically throughout the study area between 2003 and 2006, genotyped the scats to species level, and identified the prey items present in scats (see [18]). The jaguar scats were later genotyped to identify sex and individuals as part of a larger regional study of jaguar genetics [39]. In this current study, we discarded from further analyses any jaguar scats that could not be genotyped to the individual level.

### Influence of pseudo-replication in universal dietary profile

We minimised potential pseudo-replication (multiple scats originating from consumption of the same prey carcass), by excluding prey items of the same species found in scats collected on

the same date and produced by the same individual ('non-independent prey items'). In some instances, scats were collected in batches by members of the public and the precise dates and locations were unknown. In these cases, we discarded duplicate prey items from the same batch and produced by the same jaguar. From here-on, 'prey items' refers to independent prey items (each prey item represents an individual prey animal). We calculated the relative occurrence of each species in jaguar diet pooled across all individuals as (number of prey items belonging to species X)/(total number of prey items) x 100. We investigated whether the exclusion of non-independent prey items influenced the universal jaguar diet profile by comparing the results from this subset with those of our original study [18] in which we had used the dataset without identifying individuals or removing items that met the non-independent criteria.

## Sample size and dietary richness

The sample size threshold for assessing dietary richness (number of prey species in the diet) varies within and between predator species, and with the richness and availability of potential prey in the study area [24]. As such, larger samples may be required from areas of higher biodiversity, such as the equatorial tropics. Traditionally, sample size can be assessed by re-sampling the data and plotting a species accumulation curve. Sufficient samples are assumed if the number of prey species detected in the diet reaches an asymptote (no new species are detected with increasing sample size). We used rarefaction curves using the package 'vegan' in R [40–42] to explore the sample size at which the dietary richness reached an asymptote. We did this for the pooled sample (universal diet, all jaguars combined) and separately for each individual with at least three prey items. We ran the analyses using three levels of prey categorisation: (1) categorised by species (or lowest order taxa possible, N = 16 taxa); (2) six categories representing the species most commonly occurring in jaguar diet ('principle prey', all with a relative occurrence ≥5% and accounting for 85% of jaguar diet in the study area, specifically nine-banded armadillo (from here-on, 'armadillo'), white-nosed coati, white-lipped peccary, collared peccary, paca, and domestic ungulates), and a seventh category, 'rare', pooling all low incidence taxa, each with relative occurrence <5%; [18,24]; and (3) three ecological categories, representing species >10kg, armadillos, and other species ≤10kg. With the exception of one dog, we note that all prey items in the category '>10kg' (N = 58) were ungulates. Therefore, from here on we interchangeably refer to the category '>10 kg' as 'ungulates'.

## Individual variation in diet

We explored the rate at which the three ecological groups (species >10kg, armadillos, and other species ≤10kg) occurred in individual diets. For each ecological group, we calculated the number of prey items per jaguar, and regressed this against the total number of prey items (from all three groups) detected per jaguar. For each regression, we used the R-sq value to assess how much of the observed dietary variation between individuals was explained by sample size. A high R-sq would indicate that most of the variation is attributable to sample size, rather than biological differences between individuals. For ecological groups with significant regressions, we used the gradient to estimate the average dietary occurrence across individuals. For each regression, we identified individuals with large residuals (>2 standard residuals), and explored how their diet differed from the average diet. Additionally, we removed these outliers from the pooled dataset to investigate whether they influenced the relative occurrence of prey species in the universal dietary profile.

## Results

Over the 4-year period, we identified 34 jaguars from 232 scats: 27 males (219 scats), three females (8 scats) and four individuals of unknown gender (5 scats). We identified animal prey

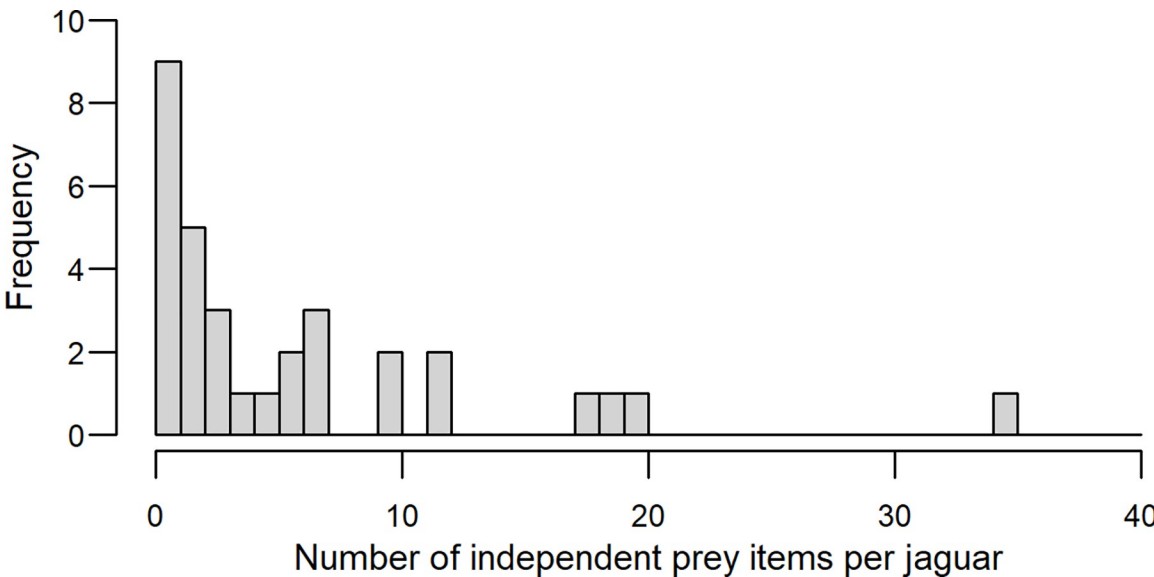

**Fig 1. Frequency distribution of prey items per jaguar.** N = 206 independent prey items from 32 jaguars; scats collected from 2003 to 2006 in and around the Cockscomb Basin Wildlife Sanctuary, Belize.

remains in 199 scats from 32 jaguars, corresponding to 176 independent scats (165 from 26 males, 7 from three females, and 4 from three jaguars of unknown gender). We detected an average of 1.2 prey items per scat (SD = 0.2, range = 1.0 to 2.0). The number of independent prey items contributed per jaguar ranged from 1 to 35 (median = 3). The dataset was skewed towards a single individual who contributed 17% (35/206) of the prey items. Eight jaguars (all male) each contributed ≥10 (≥5%) of the prey items, accounting for 66% of dataset, while 24 jaguars (18 males, 3 females, 3 unknown sex) each contributed <4 (<5%) of the prey items (Fig 1).

## Influence of pseudo-replication in universal dietary profile

Potential non-independent scats (those presumed to have originated from consumption of the same prey carcass) were identified and excluded at a rate of 11 per 100 scats. With the exception of sheep, we found no evidence that the relative occurrence of principle prey species (relative occurrence ≥5%) in pooled jaguar diet differed between the full set of scats analysed by [18] and the subset of independent scats in this re-analysis (Table 1). In the case of sheep, the relative occurrence fell minimally below the threshold of 0.05 (17/378 prey items) to 0.04 (9/ 206). This was associated with the exclusion of eight scats containing sheep remains (four that could not be genotyped to the individual level, and four non-independent scats produced by one male jaguar, M6). All other taxa identified by Foster et al. [18] as principle prey occurred with a similar frequency in the subset of independent scats as in the original study (Table 1) suggesting that, for these species, the original universal dietary profile presented by [18] was not biased by multiple scats originating from the same prey carcasses.

## Sample size and dietary richness

The number of prey taxa detected in jaguar diet did not asymptote with increasing sample size for the pooled sample of independent prey items (30 jaguars, N = 200 prey items of known taxa; Fig 2A). This can be attributed to the high number of species that occurred at a low rate,

**Table 1. Proportion of jaguars consuming each prey, and relative occurrence of prey in the pooled diet.**

| Species | | Proportion of individuals | Relative occurrence | |
|---|---|---|---|---|
| | | | Sub-sample | Full sample |
| Nine-banded armadillo* | *Dasypus novemcinctus* | 0.81 | 0.47 | 0.46 |
| White-nosed coati* | *Nasua narica* | 0.41 | 0.09 | 0.11 |
| White-lipped peccary | *Tayassu pecari* | 0.38 | 0.12 | 0.10 |
| Collared peccary | *Pecari tajacu* | 0.25 | 0.05 | 0.05 |
| Paca* | *Cuniculus paca* | 0.19 | 0.05 | 0.05 |
| Red brocket deer* | *Mazama americana* | 0.19 | 0.03 | 0.03 |
| Cattle | *Bos taurus* | 0.16 | 0.03 | 0.03 |
| Kinkajou | *Potos flavus* | 0.16 | 0.03 | 0.03 |
| Opossum | *Didelphis* sp. | 0.13 | 0.02 | 0.01 |
| Unknown mammal* | Class Mammalia | 0.13 | 0.02 | 0.01 |
| Sheep | *Ovis aries* | 0.06 | 0.04 | 0.05 |
| Green iguana* | *Iguana iguana* | 0.06 | 0.01 | 0.01 |
| Dog | *Canis familiaris* | 0.03 | <0.01 | <0.01 |
| Northern tamandua | *Tamandua mexicana* | 0.03 | <0.01 | <0.01 |
| Northern raccoon | *Procyon lotor* | 0.03 | <0.01 | <0.01 |
| Unknown ungulate | - | 0.03 | <0.01 | <0.01 |
| Unknown carnivore | Order Carnivora | 0.03 | <0.01 | 0.01 |
| Unknown rodent | Order Rodentia | 0.03 | <0.01 | <0.01 |
| Unknown bird | Class Aves | 0.03 | <0.01 | <0.01 |

Scats collected from 2003 to 2006 in the Cockscomb Basin Wildlife Sanctuary and neighbouring lands in Belize; sub-sample: 206 independent prey items from 176 scats comprising 165 from 26 males, 7 from three females, and 4 from three jaguars of unknown sex; versus full sample: 378 prey items from 322 jaguar scats, including non-independent prey items, [18]; all listed taxa were found in male scats, *indicates taxa found in female scats.

and suggests that our universal dietary profile underestimates the true dietary richness in the study area. When we pooled these rare species as a single category, 153 prey items were sufficient to detect the presence of the principle prey taxa and the group containing rare species (Fig 2B). More broadly, the three categories (ungulates, armadillos, other small vertebrates) were all detected in samples of just 48 prey items (Fig 2C). Thus, while rather large samples (> 200) would be needed to detect true richness of the pooled diet, detecting the most important species was achieved with smaller samples.

At the individual level, dietary richness did not asymptote for any of the jaguars with ≥ 3 prey items. This was expected given the low individual sample sizes (median = 7 prey items per jaguar, range = 3 to 33, N = 18 jaguars). When we pooled the rare species as a single category, the dietary richness of the jaguar with the largest sample (M1, N = 33 prey items) reached an asymptote at a sample size of 32. This jaguar's diet comprised five of the six principle prey (all except collared peccary) and rare species (specifically racoon, kinkajou, and bird). Although the low sample size for any given individual precluded robust statistical comparison of dietary richness and breadth between individuals, when we categorised more coarsely to the three ecological groups (ungulates, armadillos, and other small vertebrates), we detected all three groups in the diets of 72% (13/18) jaguars. All 18 jaguars included armadillo in their diet, and 89% (16/18) included ungulates. The two jaguars for which we did not detect ungulates each had a sample of just three prey items. The dietary richness of one individual (M16, N = 7 prey items) reached an asymptote at a sample size of 5, with no evidence of small vertebrates in his diet.

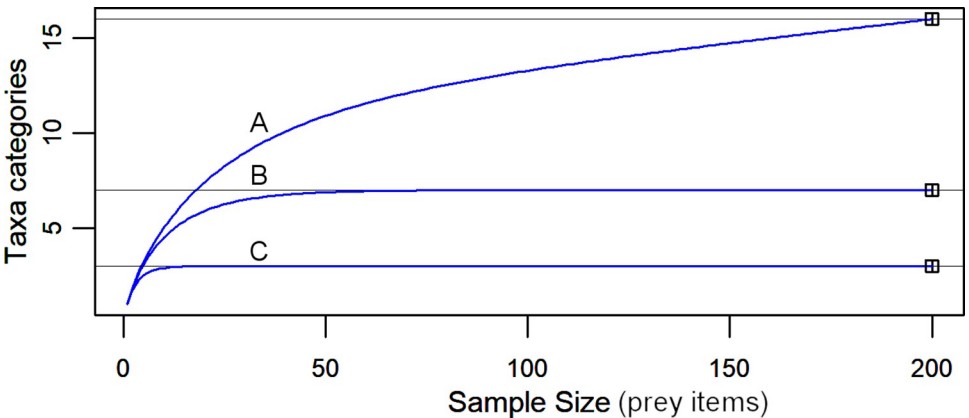

**Fig 2. Accumulation curves for pooled jaguar diet, at three levels of taxa classification.** (A) Fine-scale, 16 categories: lowest order taxa identifiable; the number of taxa does not asymptote below N = 200 prey items; (B) Medium-scale, seven categories: six principle prey groups (nine-banded armadillo, white-nosed coati, white-lipped peccary, collared peccary, paca, and domestic ungulates) and one pooled group of rare prey (each species with relative occurrence <5%); the number of taxa reaches 6.99 with N = 89 prey items, and asymptotes at 7 taxa with N = 153 prey items; (C) Broad-scale, three categories: species >10kg (ungulates), nine-banded armadillo, and other species ≤10kg; the number of taxa reaches 2.99 with N = 18 prey items, and asymptotes at 3 taxa with N = 48 prey items. N = 200 independent prey items of 16 known taxa from 30 jaguars; scats collected from 2003 to 2006 in and around the Cockscomb Basin Wildlife Sanctuary, Belize.

## Individual variation in diet

For male jaguars, we detected 16 prey taxa (N = 165 scats), while for the smaller sample (N = 7 scats) from female jaguars, we detected five prey taxa, all of which were present in male diet (Table 1). The maximum number of prey taxa detected in the diet of a single individual was 10 for a male (N = 14 scats) and five for a female (N = 4 scats). This remarkably high species richness from so few female scats is of anecdotal interest, but not of statistical significance.

The majority of sampled jaguars (81%, 26/32) ate armadillos, two-thirds ate large vertebrates (ungulates, 21/32) and 41% (13/32) ate white-nosed coati, the most commonly-eaten small vertebrate (sample size range = 1 to 35 prey items per jaguar; Table 1). Approximately half of the jaguars (17/32) ate peccaries and, notably, all were male: 46% of males (12/26) ate white-lipped peccary and 31% (8/26) ate collared peccary. Likewise, approximately one-fifth of males (5/26) were found to have consumed non-wild species, specifically sheep, cow and dog (Table 1). All domestic ungulate remains were from scats found in the fragmented, unprotected landscape. Closer inspection revealed that half (5/9) of the jaguars sampled in these lands had eaten domestic ungulates (sheep and/or cattle). Approximately two-thirds (69%, 11/16) of the domestic ungulate remains were associated with one jaguar (M6). For scats found inside contiguous forest, only one out of 31 jaguars had non-wild animals (dog) in their diet. Domestic dogs from neighbouring villages occasionally roam into the forest block.

We assessed whether the three prey groups, vertebrates >10kg (ungulates), armadillos, and other vertebrates ≤10kg, each occurred at a similar rate across the diets of all sampled individuals, and how much of the variation between individuals could be attributed to their sample sizes. For each group, we found a significant positive linear relationship between the frequency of occurrence (number of prey items of group X) and the sample size (vertebrates >10kg: $y = 0.24x + 0.32$, $F_{1, 28} = 24$, $p < 0.001$, R-sq = 46%; armadillos: $y = 0.53x – 0.31$, $F_{1, 28} = 201$, $p < 0.001$, R-sq = 87%; and vertebrates ≤10kg: $y = 0.23x - 0.01$, $F_{1, 28} = 72$, $p < 0.001$, R-sq = 71%; Fig 3). Overall, for the diet of an average jaguar, armadillos comprised ~53% of prey items, ungulates comprised ~24% of prey items, and small vertebrates comprised ~23% of

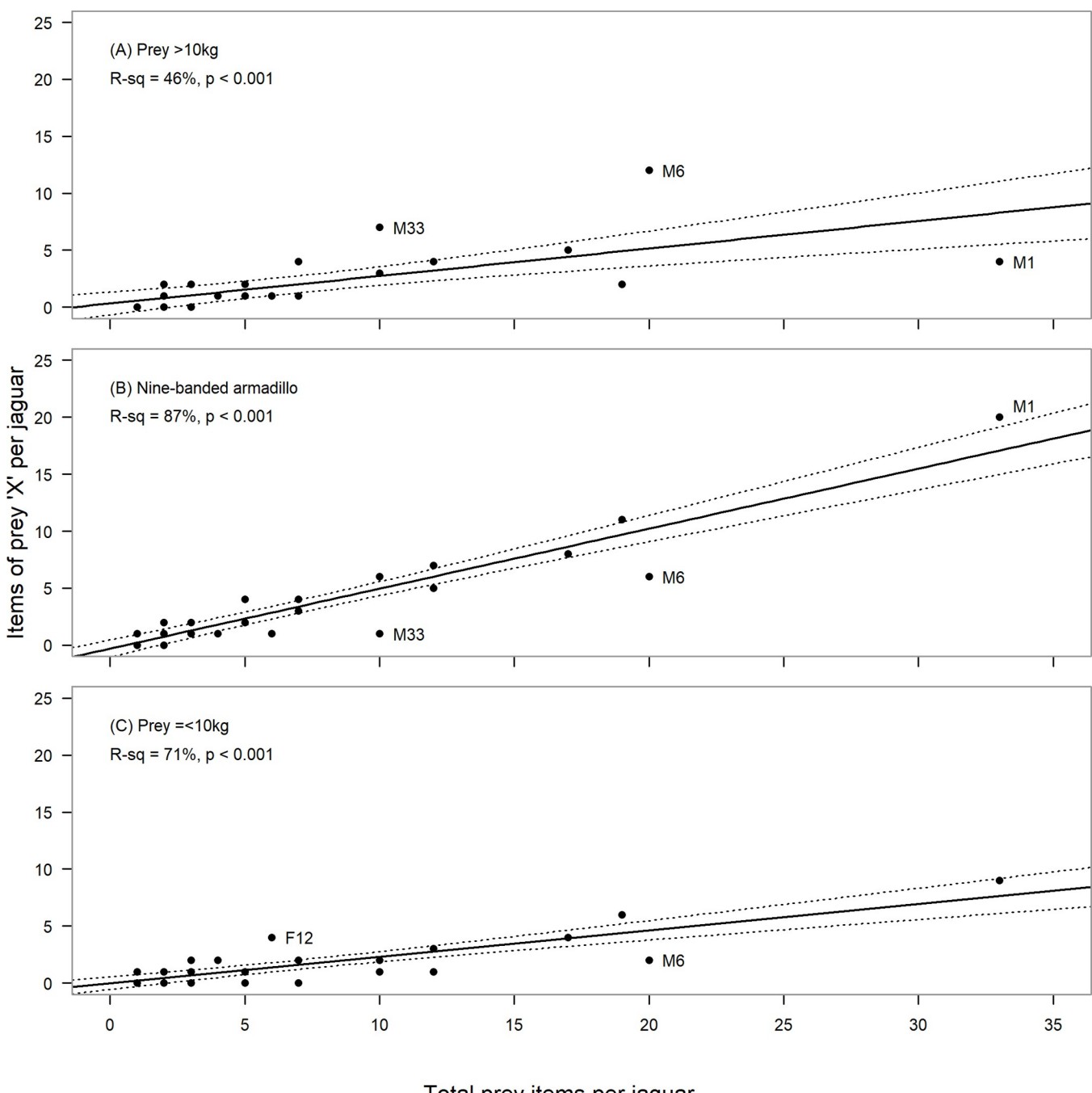

**Fig 3. Variation in number of taxa 'X' items with sample size in diets of jaguars.** Taxa X represents A) prey items >10kg, specifically white-lipped peccaries, collared peccaries, red brocket deer, cattle, sheep and one dog, B) nine-banded armadillos, and C) other prey items ≤10 kg, nine taxa. Each observation represents a unique jaguar sampled by collection of scats in the Cockscomb Basin Wildlife Sanctuary and neighbouring unprotected lands in Belize from 2003 to 2006 (30 jaguars, 200 independent prey items for which a weight class could be assigned; jaguars with large standardised residuals (>2) are labelled, M = male, F = female); 95CI shown as dotted line.

prey items. For both armadillos and other small prey, most of the observed variation between individuals was explained by their sample sizes (87% and 71% respectively, Fig 3), suggesting that the majority of jaguar individuals more or less followed the generalised jaguar diet. For ungulates, only half of the variation (46%) was attributed to sample size. Thus, while sample

size bias explains much of the variation between individuals in the consumption of small prey and armadillos, it explains less of the variation in the consumption of ungulates, suggesting that biological/ecological factors may be more relevant here.

Across the three regression analyses, we noted large standardised residuals (>2) for four jaguars, suggesting that they were not behaving in the same way as the 'average' jaguar. Specifically, for the male jaguar contributing the most scats to the dataset (M1), the occurrence of large prey (ungulates) was lower than average (Fig 3A), while the occurrence of armadillos was higher than average (Fig 3B). For this jaguar, the majority of prey items (82%, 26/32) came from the forest block and armadillos accounted for 77%. For two male jaguars (M33 and M6), the occurrence of large prey (ungulates) was higher than average (Fig 3A), and the occurrence of armadillos lower than average (Fig 3B). For M33, all 10 prey items (8 scats) were found in the protected forest, and 60% of the prey items were white-lipped or collared peccaries. For M6, all 17 scats (20 prey items) were found in the fragmented landscape, with the majority (14 scats; 16 prey items) from farmland. Overall domestic ungulates comprised 55% of his prey items. Additionally, M6 had fewer than average small prey items (≤10kg) in his diet (Fig 3C). In contrast, a female (F12) had more small prey items in her diet than average (Fig 3C), and all originated from farmland (4 scats, 6 prey items).

Compared with the other prey species, armadillo, white-lipped peccary and sheep all had a notably high occurrence in the pooled jaguar diet given the number of individuals detected eating them (Table 1). Three individuals, M1, M6, and M33 respectively contributed more armadillos, domestic ungulates, and peccaries than expected (Fig 3). Removing these three males from the pooled dataset had no influence on the relative occurrence of armadillos (0.47, N = 32 jaguars to 0.49, N = 29) or white-lipped peccaries (0.12 to 0.13), or collared peccaries (0.05 to 0.05); but excluded sheep entirely from the universal dietary profile (0.04 to 0.00) and reduced the relative occurrence of cows (0.03 to 0.02). Sheep were not detected in the diet of M33 and comprised only 3% (1/35 prey items) of M1s diet, but 40% (11/20) for M6, suggesting that M6 commonly encountered sheep within his range, or may have been specializing on sheep.

## Discussion

Our sample of jaguar scats, collected opportunistically during 4 years of continuous fieldwork in moist tropical forest and surrounding farmlands in Belize, is the largest sample used for describing jaguar diet to date (S1 Table). We surveyed an area with one of the highest recorded densities of jaguars in the region [38]. Accordingly, the number of individuals represented by this sample is high when compared with other dietary studies that have identified individual genotypes from scats (32 jaguars, this study; versus 3 to 16 jaguars, S1 Table). In our study, approximately half of the 322 scats either could not be individually genotyped or represented non-independent prey, leaving a useable sample of 176 independent scats (206 prey items). This (pooled) sample likely underestimated the true dietary richness, but was large enough to have detected any prey group occurring in the diet at a rate of ≥5%. Thus, we are confident that the universal dietary profile describes the most important prey taken. With half of the jaguars each represented by less than four prey items, we could not reliably compare individual diets. Although we could not assess species-level richness and breadth for individuals, we investigated individual dietary richness at a coarser level, assessing the relative importance of large vertebrates (primarily ungulates), nine-banded armadillos, and other small vertebrates.

The jaguars sampled in our study area almost universally ate armadillos. No other prey species were eaten by such a high proportion of the sampled population (81%), with the next three most widely-eaten species detected in the diet of only one-half to one-quarter of the

jaguars. Not only were armadillos eaten by the majority of jaguars, but they also formed a significant dietary component, averaging just over one-half of a jaguar's prey items. The remainder of an average jaguar's diet was equally spread between vertebrates larger than 10kg (ungulates, primarily peccaries), and species that are similar or smaller in size than armadillos (primarily coatis and pacas). Almost two-thirds of the consumed ungulates were peccaries, and they were eaten by half of the sampled jaguars. Indeed, peccaries have been documented as key prey items at sites across the jaguar range from Mexico to southern Brazil (e.g. Mexico [43–46], Costa Rica [47,48], Peru [49], Paraguay [50], Brazil [9,31,51,52]). Coatis, which comprised the majority of the non-armadillo small prey, were consumed by almost one-half (41%) of the sampled jaguars, and have been noted as common prey items in jaguar diet at sites in Mexico, Guatemala and Brazil [3,28,30–32].

The Cockscomb Basin jaguar population lives at high density with a stable resident population characterised by multiple overlapping ranges [37,38,53,54]. It is notable that they have existed on a relatively narrow, armadillo-rich diet for at least 20 years [18,55]. The ability of these jaguars to make a successful living from a diet in which three-quarters of their prey items are 10kg or less may be attributed partly to their small body size compared to other jaguar populations [55–57]. Potentially, dependence on relatively small prey has selected for smaller body size in this region.

The predominance of armadillos in the diets of the Cockscomb jaguars likely reflects a locally abundant armadillo population sustained by the mesic conditions of the study area [3,18]. We hypothesise that the high density and spatial overlap observed in this jaguar population [37,38,53,54] is contingent on hunting a prolific base of armadillos. Armadillos have been highlighted as a key prey species for many top and meso-carnivores [58], exhibiting several species-specific characteristics that make them a particularly suitable prey. Armadillos are a fat-rich food source, with a higher energetic content than either coatis or ungulates such as white-tailed deer [59]. Their relatively small body size, solitary existence, nocturnal behaviour, poor eyesight, and noisy foraging strategy make them easy prey for jaguars [60–63]. In contrast, the next three most commonly-eaten prey, white-lipped and collared peccaries, and white-nosed coatis, are diurnal, group-living species, which avoid predation through shared vigilance and the use of alarm calls, mobbing, attack behaviour, and/or rapid flight response [64–66]. Tackling large and/or group-living species comes with an associated risk of injury and failure, and in the case of large prey such as peccaries or domestic ungulates, an extended time investment of staying with the kill over several days to maximise use of the carcass. Forming the core of jaguar diet throughout the study area, we infer that armadillos are homogenously distributed across the landscape. In contrast, large ungulates are generally more heterogeneously distributed in time and space; in particular white-lipped peccaries which must be followed over large distances, ranging widely and seasonally in herds of up to 200 individuals [61,62,67,68] With smaller and potentially more predictable home ranges, and reproductive rates three to four times as high as those of peccaries [69–72], armadillos likely are a more reliable prey than peccaries. Where armadillos occur at high density, their small size, high energy content and relative ease of capture potentially support high jaguar density, facilitating mobility and overlapping ranges. Exploiting abundant armadillos while 'on-the-go' may be a particularly useful foraging strategy for males, allowing more time to traverse the landscape searching for females, defending ranges and deterring competitors. The jaguar for whom we found the most scats, a male, contributed 48 scats over 3.5 years throughout the western section of the study area, indicative of stable range use. Notably, armadillos occurred at an above-average level in his diet, accounting for 60% of his prey items. Depositing scats in conspicuous places, such as trail systems, is common within big cats and may indicate residency or dominance in an area [54,73,74]. Given the large number of scats found over a

prolonged period for this male, we indirectly infer that this cat was resident, potentially dominant, in this part of the study area. We hypothesise that his apparent success is associated with exploiting an abundant armadillo resource, enabling him to travel rapidly between meals for efficient area coverage.

Although armadillos potentially function as handy, calorie-dense, snack-packs for jaguars, energetic models suggest that consumption of armadillos or other small prey is limited by the high kill rates that would be necessary if larger prey were not also taken [3,18,35]. Our study supports this premise, with large ungulates detected in the diets of the majority of jaguars. Furthermore, we found evidence of dietary shifts within individuals, between large native species and livestock. The diets of five males, whose ranges spanned the boundary of the protected area, included wild and domestic ungulates. Notably, the wild ungulates were present in scats only found inside the Cockscomb Basin Wildlife Sanctuary and the domestic ungulates in scats found on farmland. This suggests opportunistic foraging behaviour, switching between wild and domestic ungulates according to the local availability. Similarly, in the Pantanal, Brazil, Cavalcanti & Gese [9] found that jaguars commonly switched between native prey and cattle, and explained this as a flexible response to temporally and spatially varying food resources. In our study, one male had a markedly high occurrence of sheep and cattle in his diet. However, without detailed data on prey encounter rates, we cannot discern whether this individual was specialising on livestock. Across the same study area, Rabinowitz [75] found evidence that infirm jaguars were associated with depredation, whereas Foster [35] noted that apparently healthy jaguars, of both sexes, also attacked livestock. While old age or injury may be an important driver of movement to marginal lands, where the encounter rate with 'easy' domestic prey may be higher, this does not preclude healthy individuals from opportunistically killing livestock or establishing ranges around these available resources. In this study, over half (5/9) of the jaguars sampled in the unprotected fragmented landscape ate domestic animals. This suggests that livestock depredation is a relatively widespread behaviour among the jaguars in the local area. The remaining jaguars sampled in this human-influenced landscape (4/9) showed no evidence of livestock in their diets. While this absence may be an artefact of the small sample sizes for these individuals (1–6 prey items per jaguar), it indicates that jaguars still successfully find wild prey when traversing this area.

Sex-based differences in diet have been documented in felid species (e.g. jaguars, [9]; leopards, [12,13,76]; cheetahs, *Acinonyx jubatus*, [77]; pumas [78]). These differences have been associated with differences in body size and reproductive status, with males taking larger prey than females, and females with dependents taking larger prey than solitary females [13,76–78]. In the case of jaguars, Cavalcanti & Gese [9] found that male jaguars killed a higher proportion of peccaries than did females, while females killed a higher proportion of caiman (*Caiman crocodilus*) than did males. They suggested that these differences are associated with male-female differences in mobility and body size, making it easier for males to follow and attack wide-ranging herds of white-lipped peccaries. Our samples from females were too few to rigorously explore sex-based differences in jaguar diet and also biased towards the unprotected fragmented landscape (6/8 prey items). However, in the female scats we detected no peccary, and only one large vertebrate prey item, red brocket deer, a solitary forest ungulate. The absence of peccaries is noteworthy, but requires a larger sample size from females to make inferences about differences in species consumption between the sexes.

Adequate sample size, distributed evenly across a representative number of individuals, is necessary for describing universal and individual dietary profiles. Although the current study possibly represents one of the best scat datasets available for analysing jaguar diet to date, it is skewed by the high number of individuals with low sample sizes and by an over-representation of males versus females (individuals and sample sizes). In this study, 94% of scats originated

from male jaguars, effectively restricting our conclusions about jaguar diet to males only. Under-sampling of females is common during surveys for jaguar scats, with more male jaguars detected than female jaguars, and more scats found per male than per female (this study, [26,43,56,79]; but see [36]). We found 165 scats from 26 males, and only seven scats from three females. In our study area, the low number of female scats versus male scats found along trails mirrors detection rates by trail-based camera traps [37,80]. Given that the local population sex ratio is considered relatively even [37], we attribute this male-bias in scats to large-scale differences in ranging behaviour between the sexes and to differential trail use and marking behaviour. Female jaguars are philopatric, whereas males disperse widely and maintain larger ranges, overlapping with both sexes [54,56,73,81,82]. Under such a system, we expect to detect more males than females as males move in and out of the survey area, while the resident females remain [53]. Additionally, males dominate the trail systems and defecate in conspicuous areas, while females avoid trails frequently traversed by males, except when receptive to breeding [53,80]. Thus, we expect fewer female than male scats are deposited along trails. Given the high likelihood of under-sampling female jaguars during trail-based scat surveys, and the possibility that their diet may differ from that of males, we recommend genotyping samples at least to level of sex for dietary studies, allowing for an assessment of potential sex-based bias in the dataset.

Because of the low rate of finding jaguar scats, and the considerable effort required, there may be a general reluctance among researchers to discard duplicate samples from diet analyses. However, the inclusion of non-independent scats and the over-representation of a few individuals in the dataset can lead to misleading dietary profiles, particularly in small samples. In our study, the contribution of non-independent scats marginally influenced our original interpretation of the universal dietary profile in which sheep was documented as a principle prey species [18,24] and later found to have been eaten by just two out of 32 sampled jaguars. The extent to which this phenomenon is a problem when interpreting jaguar scat data is unknown, but may be common. For example, Gonzalez-Maya et al. [83] concluded that ocelots are a seasonally important prey item for jaguars in the mountains of Costa Rica. They based this conclusion on 15 scats collected over 7 months. One-third of the scats contained evidence of ocelot. However, without information on the spatial or temporal distribution of the scats, we cannot discount the possibility that these five scats originated from a single predator and from the same predation event. The probability of surveyors finding multiple scats originating from the same cat and same prey carcass will depend partly on the species consumed, for example, if the carcass is large and provides several meals, or contains a high proportion of indigestible material. Precise information on the time and location of scat deposition, and ideally the individual that produced it, will help when deciding whether to discard scats that contain the same prey species. The estimated date of defecation can be narrowed down by surveying frequently and assessing the relative freshness of the scats. Reliable records of the collection location will allow accurate estimates of the distances between collected scats, and thus the likelihood that they were deposited by the same individual. This may be augmented with camera-trap or tracking data revealing the number and identities of jaguars detected on different sections of the trail system. Genotyping of scats to the level of sex or individual, if possible, will provide still further evidence to assess independence.

In this study, the majority of scats (two-thirds) came from just one-quarter of the sampled individuals (8/32 jaguars). A left-skew in the distribution of sample sizes across individuals may be problematic for describing the universal diet in populations where individual diets differ significantly. If the diets of individuals contributing more scats differ from those contributing few scats, then the universal dietary profile may not adequately represent the overall feeding habits of the population. In this study, we found that virtually all the sampled

individuals followed the same pattern of a 2:1:1 ratio of armadillos, large vertebrates (ungulates) and small vertebrates in their diets; and removing the few outliers did not change the overall universal dietary profile. The left-skew in the distribution of sample sizes across individuals likely reflects the spatial and temporal survey effort relative to the range size and movement patterns of local jaguars. If one or a few individuals dominate the survey in space and/or through time we would expect them to provide the majority of scats, with fewer scats originating from multiple individuals who overlapped minimally with the survey area, trail system and/or sampling period. In the few scat-based studies of jaguar diet that have genotyped individuals, the individual sample sizes are not reported [36,84,85]. Where such data are available we encourage researchers to report the frequency distribution of scats across individuals so that the worth of the universal dietary profile may be evaluated.

To improve our understanding of female jaguars, and further explore generalist and specialist feeding behaviour at the individual level, future scat-based studies of individual jaguar diet will need larger sample sizes. If the cost and logistics are not prohibitive, sample size could be increased by extending the survey period or survey area. Expanding the survey area, relative to home range size, or searching more varied locations within the survey area (e.g. off-trail as well as on-trail), may increase the number of individuals sampled, but not necessarily the number of samples per individual (e.g. this study versus [79,86]). Conversely, extending the survey period may increase the number of samples from the same individuals, improving the accuracy of individual diet profiles; and allow the tracking of longitudinal changes in an individual's diet associated with phenotypic or environmental changes through time. Alternatively, detection rates of scats could be boosted with the use of detector dogs. Detector dogs are widely used to detect elusive species for field research [87,88]. While there have been few formal standardised comparisons of dogs and human surveyors [87], there is growing evidence that trained dogs are better than humans at finding carnivore scats (e.g. brown bears, *Ursos arctos*, [89]; mustelids, [90]). As part of a felid genetics study, Wultsch et al. [79,86] successfully used a detector dog to find jaguar scats on- and off-trail across a range of neotropical habitats. Sampling 16 sites across Belize over a 4-year period, they found 299 jaguar scats representing 65 jaguars (a similar number of scats as this study, but twice as many individuals). Despite this impressive survey effort and sample collection, it averages only 4.6 scats per jaguar. Even with intensive search effort, sample sizes remain on the low side for adequately describing individual diet. This is the reality of scat-based dietary studies of elusive, wide-raging, forest-dwelling carnivores such as jaguars. For the majority of jaguar diet studies, scat collections are the norm, and limited sample sizes are inevitable. In the absence of anything better, researchers need to make use of the available scat data while taking care not to overinterpret the results given the biases that may arise within small and skewed samples.

## Conclusions and implications

Our study has shown that our previously published universal dietary profile of jaguars in and around the Cockscomb Basin in Belize [18] came from a sample of males which all ate the same broad prey categories in a similar ratio. Considering the widespread bias in finding male versus female scats [26], it seems likely than many of the existing scat studies to date reflect the diet of male jaguars. Although we have a good understanding of male diet, we need more information about the diet of females, as the sex responsible for raising off-spring and driving population change. A reduction in the availability of prey for female carnivores negatively affects recruitment [91]. To identify and address this potential threat in jaguar populations, for example where wild prey are being overharvested by humans (e.g. [36,69], we need more data on female dietary needs from across the jaguar range. Overall, our study has revealed a

consistent but narrow diet throughout the (male) jaguars of the Cockscomb Basin, and has highlighted the importance of armadillos and peccaries in supporting this high-density population. Little is known about peccary densities throughout their neotropical range, and even less so about nine-banded armadillos. Changes in the availability of these ecologically different prey species will likely influence the social system, ranging patterns and density of jaguars in a complex and interactive way. We recommend research into the relationship between jaguar density and the distribution and abundance of armadillos and peccaries, to understand the regulatory role of these quite different prey species in the population dynamics of the largest cat of the Neotropics.

Sampling protocols and sample sizes are a major concern for the reliable interpretation of many studies of jaguar diet to date. To overcome these issues in future studies, we recommend genotyping to the level of species, at least, if multiple carnivore species exist in the study area. When possible, we recommend that the scats are further genotyped to identify sex and individual. This will facilitate the identification and removal of non-independent scats, and enable an assessment of any skew in the dataset between the sexes or across individuals. In the absence of genetic data, researchers could consider the use of other data which may help identify the individual who produced the scat (e.g. camera-trap photos, GPS tracking locations). We also recommend the use of species accumulation curves to assess whether sample sizes are large enough to describe dietary richness. When sample sizes are insufficient, researchers should consider pooling species into ecological groups for making inferences about the consumption of prey taxa at a broader scale than species level. Resources permitting, sample sizes could be boosted by extending the survey period or expanding survey area, by increasing the number of surveyors or making use of scat detector dogs.

## Supporting information

**S1 Table. Species identification and sample size in scat-based studies of jaguar diet.** Scat-based studies of jaguar diet from published and unpublished sources, showing the study country, the number of years during which scat collection was conducted, the method for assigning felid species to the scats ('signs' includes tracks, scrapes, and resting places), whether the scats were genotyped to the level of individual jaguar, whether criteria were applied to exclude potentially non-independent scats (those presumed to have been produced by the same individual feeding from the same carcass), and the number of scats that were ultimately analysed to describe the diet.
(DOCX)

**S1 Data. Number of independent prey items per jaguar.** Dataset used for Fig 1.
(XLSX)

**S2 Data. Number of independent items of each unambiguous prey taxa for which a weight class could be assigned, per individual jaguar.** Dataset used for *Sample size and dietary richness* (Fig 2) and *Individual variation in diet* (Fig 3).
(XLSX)

## Acknowledgments

We thank the Belize Audubon Society for facilitating our research in the Cockscomb Basin Wildlife Sanctuary for almost two decades, and the Government of Belize's Forest Department for supporting our long-term work in Belize. For the genetic analysis of scat samples, we thank Claudia Wultsch and the Sackler Institute for Comparative Genomics at the American Museum of Natural History. We also thank Patrick Doncaster at the University of

Southampton for his academic support over the years. This work is dedicated posthumously to Dr Howard Quigley who left us too soon. We want thank him for his continued mentorship and encouragement of our research. His passion for wild cats will continue to inspire researchers, conservationists, and naturalists, for years to come.

## Author Contributions

**Data curation:** Rebecca J. Foster.

**Formal analysis:** Rebecca J. Foster.

**Investigation:** Rebecca J. Foster.

**Methodology:** Rebecca J. Foster, Bart J. Harmsen.

**Project administration:** Rebecca J. Foster.

**Writing – original draft:** Rebecca J. Foster, Bart J. Harmsen.

**Writing – review & editing:** Rebecca J. Foster, Bart J. Harmsen.

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
