## [Decision Letter · Decision Letter 0]

20 Apr 2022

PONE-D-21-29787Dietary similarity among males in a high-density jaguar (*Panthera onca*) populationPLOS ONE

Dear Rebecca Jecquueline Foster

Thank you for submitting your manuscript to PLOS ONE. After careful consideration, we feel that it has merit but does not fully meet PLOS ONE’s publication criteria as it currently stands. Therefore, we invite you to submit a revised version of the manuscript that addresses the points raised during the review process.

We look forward to receiving your revised manuscript.

Kind regards,

Bilal Habib

Academic Editor

PLOS ONE

Journal Requirements:

2. Thank you for stating the following financial disclosure: "RJF and BJH were funded through Panthera (www.panthera.org) by the Sitka Foundation (www.sitkafoundation.org) and the Liz Claiborne Art Ortenberg Foundation (www.lcaof.org). The funders had no role in study design, data collection and analysis, decision to publish, or preparation of the manuscript."

We note that one or more of the authors is affiliated with the funding organization, indicating the funder may have had some role in the design, data collection, analysis or preparation of your manuscript for publication; in other words, the funder played an indirect role through the participation of the co-authors. If the funding organization did not play a role in the study design, data collection and analysis, decision to publish, or preparation of the manuscript and only provided financial support in the form of authors' salaries and/or research materials, please do the following:

a. Review your statements relating to the author contributions, and ensure you have specifically and accurately indicated the role(s) that these authors had in your study. These amendments should be made in the online form.

b. Confirm in your cover letter that you agree with the following statement, and we will change the online submission form on your behalf: 

“The funder provided support in the form of salaries for authors [insert relevant initials], but did not have any additional role in the study design, data collection and analysis, decision to publish, or preparation of the manuscript. The specific roles of these authors are articulated in the ‘author contributions’ section.

3. We noted in your submission details that a portion of your manuscript may have been presented or published elsewhere. "Foster et al 2010, Journal of Zoology 280: 309-318' Please clarify whether this publication was peer-reviewed and formally published. If this work was previously peer-reviewed and published, in the cover letter please provide the reason that this work does not constitute dual publication and should be included in the current manuscript.

Additional Editor Comments (if provided):

The reviewer has highlighted major issues in this paper. I request authors to change as suggested by the reviewer. As an associate editor i apologies for the delay. The delay was because on no response from reviewers.

Reviewers' comments:

Reviewer's Responses to Questions

**Comments to the Author**

1. Is the manuscript technically sound, and do the data support the conclusions?

Reviewer #1: Yes

2. Has the statistical analysis been performed appropriately and rigorously? 

Reviewer #1: Yes

3. Have the authors made all data underlying the findings in their manuscript fully available?

Reviewer #1: Yes

4. Is the manuscript presented in an intelligible fashion and written in standard English?

Reviewer #1: No

5. Review Comments to the Author

Reviewer #1: This is very important paper in not only Jaguar feeding ecology but also developing ideas around individual diet variation in carnivores. The analytical methods are excellent and results are repeatable. The paper, however, suffers from the lack of a systematic sampling design which brings in strong biases in the results. I appreciate that the authors are clearly aware of the limitations and have interpreted their results in the right spirit.

Following are my main concerns on this paper:

1) This research did not start with the investigation of male only diet, which is suggested by the current title of the paper. The mismatch continues in the Introduction, which is developed around the need for profiling individual-level diets of jaguar. The hypotheses and objectives in the introduction do not indicate that this is going to be a research of primarily male diet. Instead sex-based variations seem to be an interest of the authors initially. The situation changes when the authors found out that their results are limited to mostly male diet after genotyping the scats.

Hence I suggest to change the title to reflect it correctly that limiting to male diets was due to the results and not a main objective of this paper. Alternatively, a lot of changes will be required in the Introduction and I don’t see a clear direction there. Additionally, the authors could establish in the Introduction that sex-based diet variations were expected, and indicate to the readers that this paper could not cover both the sexes to draw meaningful conclusion.

2) Since the term ‘prey item’ is also used in the literature to denote the prey species, it could cause some confusion and need to be clarified in the abstract as well (line 20).

3) Neither the test applied nor test test-statistics to measure the difference between average prey items per scat for males and females are mentioned (line 239). Also, because there were only 7 scats from 3 females, any such analysis won’t be meaningful.

4) Since sample size explained most of the diet diversity (line 330 to 335), these results best stand as suggestive. However, there is learning in knowing that we haven’t reached sufficient sample size.

5) Paragraph from line 459 to 479 can be shortened as there isn’t enough data to either support or reject any hypotheses on sex-based diet variations.

6) Line 491-492: Differential trail-use alone doesn’t seems to be sufficient explanation for such huge sampling bias towards male. Is there any evidence of the population itself being skewed towards one sex?

7) Line 590-592: Does the sampling scheme and sampling size of the old studies allow for comparison with this study and derive a conclusion that jaguar diets have changed? Your paper strongly argues that the studies with small sample size and without individual-level identification may have inherently skewed results.

8) Conclusion section should have only essential references. Arguments and references as supporting evidence should go to the discussion (e.g. line 626-629). Overall, the conclusion section is quite lengthy and can be shortened to keep it focused only on major points and recommendations.

9) A line about sampling recommendation in overall conclusion will add value. A big take home message of this paper is that sampling is a major issue at hand for most studies of jaguar diet.

Minor fixes:

Line 39: correct scientific name of leopard should be Panthera pardus.

Line 57: Mention of full scientific name (Panthera pardus) second time could be omitted

Line 75: Better to mention the authors by name- 'de Oliveria [22] noted...'. Makes it easy to read.

Line 155: ‘this’ is repeated

Line 514: ‘information one the’ to ‘information on the’

6. PLOS authors have the option to publish the peer review history of their article (what does this mean?). If published, this will include your full peer review and any attached files.

Reviewer #1: No

---

## [Author Response · Author response to Decision Letter 0]

4 Jun 2022

Dear Dr Habib

We thank you and the Reviewer for the constructive comments on our manuscript entitled ‘Diet similarity among jaguars (Panthera onca) in a high-density population’.

We confirm that we agree with the following statement – “The funder provided support in the form of salaries for authors [RJF and BJH], but did not have any additional role in the study design, data collection and analysis, decision to publish, or preparation of the manuscript. The specific roles of these authors are articulated in the ‘author contributions’ section.”

This study builds on our earlier peer-reviewed and formally published work on jaguar diet (Foster et al 2010, Journal of Zoology 280: 309-318). This work does not constitute dual publication and should be included in the current manuscript because it represents scats genotyped only to the level of species, presenting a universal diet pooled across an unknown number of jaguars. In our current study (this submission), the same scats were genotyped to the level of the individual, allowing for the detection and assessment of pseudo-replication in the universal diet described by Foster et al 2010. Therefore, the inclusion of the 2010 results in this submission is for comparison only and does not constitute dual publication. They are displayed and properly referenced in Table 1.

Please see our responses to the Reviewer below:

Major fixes (line numbers refer to the marked up/tracked changes version)

(1) We have changed the title to ‘Diet similarity among jaguars (Panthera onca) in a high-density population’. We have deleted the sentence that implies that we will be investigating sex-based differences (lines 149). We have added the following ‘As long-term camera trap data from the study site revealed male-biased use of the trail system (Harmsen et al 2017), we anticipated a male bias in the scat data also.’ (lines 156-157).

(2) To clarify that ‘prey items’ refers to individuals not species, we have added ‘individual prey animals’ in parentheses after our first reference to ‘prey items’ in the Abstract (lines 21-22). 

(3) We have removed the average prey item per scat for males and females, and replaced it with the average number of prey items per scat for jaguars; and we have included the standard deviation and range (lines 242-243). 

(4) The Reviewer agrees that these results are suggestive and does not request we make any specific changes to the text; indeed, we already indicate that the results in this section are suggestive. To further clarify, we have rewritten the final sentence to make it clearer that while sample size bias explains much of the variation between individuals in the consumption of small prey and armadillos, it explains only a low amount of variation between individuals in the consumption rate of ungulates, suggesting that biological/ecological factors may be more relevant here (lines 347-351). Additionally, we have explicitly stated the sample sizes at which prey taxa reached an asymptote for each of the three scales of prey categorization (Figure 2, lines 288-298).

(5) We have shortened the paragraph about sex-based diet variations (lines 498-504).

(6) Evidence (camera-trap data) from the same study area shows that females only use the trail system sporadically and have lower detection probability than males, however there is no evidence of the population being skewed towards one sex, with long-term monitoring revealing a relatively even sex ratio (Harmsen et al 2017, Harmsen et al 2020). Therefore, as sampling for scats only occurred along trails, we consider that differential trail-use is a sufficient explanation for the male sampling-bias. We have rewritten the sentence to indicate that the sex ratio is not skewed (line 526).

(7) We have removed from the Conclusion that jaguar diet has changed in the study area. We have noted in the Discussion that armadillos have remained a core component of jaguar diet over a period of 20 years, citing Rabinowitz & Nottingham (1986), a study from the same study area, which has the third largest sample of jaguar scats to date in the literature (228 scats, Table S1), comparable in size to the number of scats used in this current study (322 scats), and considered large enough to reliably detect the principle prey taxa.

(8) We have shortened the Conclusion (deleted lines 643-701), and rewritten and moved arguments and references as supporting evidence to the Discussion (lines 406-453, 611-627).

(9) We have included some sampling recommendations in the Conclusion (lines 628-642).

Minor fixes

(1) We have corrected the scientific name of leopard 

(2) We have deleted (Panthera pardus) 

(3) Where referring to particular studies, we have mentioned the authors by name throughout 

(4) We have deleted the repeated ‘this’ 

(5) We have changed ‘information one the’ to ‘information on the’

We believe that we have fully responded to all your comments. We look forward to hearing from you. 

Yours faithfully

Rebecca Foster, PhD

References

Harmsen BJ, Foster RJ, Sanchez E, Gutierrez-Gonzalez CE, Silver SC, Ostro LET, et al. Long term monitoring of jaguars in the Cockscomb Basin Wildlife Sanctuary, Belize; Implications for camera trap studies of carnivores. Plos One. 2017;12(6). doi: 10.1371/journal.pone.0179505. PubMed PMID: WOS:000404607900028.

Harmsen BJ, Foster RJ, Quigley H. Spatially explicit capture recapture density estimates: Robustness, accuracy and precision in a long-term study of jaguars (Panthera onca). Plos One. 2020;15(6):19. doi: 10.1371/journal.pone.0227468. PubMed PMID: WOS:000540938100031.

Rabinowitz AR, Nottingham BG. Ecology and behaviour of the jaguar (Panthera onca) in Belize, Central America. Journal of Zoology. 1986;210:149-59.

---

## [Editor Report · Decision Letter 1]

2 Aug 2022

PONE-D-21-29787R1Dietary similarity among jaguars (*Panthera onca*) in a high-density populationPLOS ONE

Dear Dr. Foster,

Thank you for submitting your manuscript to PLOS ONE. After careful consideration, we feel that it has merit but does not fully meet PLOS ONE’s publication criteria as it currently stands. Therefore, we invite you to submit a revised version of the manuscript that addresses the points raised during the review process.

We look forward to receiving your revised manuscript.

Kind regards,

Bogdan Cristescu

Academic Editor

PLOS ONE

Journal Requirements:

Additional Academic Editor (Bogdan Cristescu) Comments (if provided):

This is an interesting manuscript that has already gone through a round of revisions. I only have minor comments below. Line numbers pertain to the manuscript with track changes.

L158: insert "is" after "this"

L637: "sample" instead of "samples"

L315-317: did you discuss that this result might be a reflection of smaller sample size for females than for males?

Question: Did the scats confirmed as jaguar originate from adults, subadults, kittens also? The inclusion of scats from young jaguars in the analysis could provide partial explanation of the high taxonomic richness of prey items in the scat of male jaguars. This is something to consider adding to the Discussion

Fig. 1: why not standardize the data by dividing number of prey items by number of scats processed for each jaguar?

Fig. 2: add "scats" in brackets

Fig. 3: if you chose to include the jaguar ID for the data that fall outside the confidence interval bands, why not include it for all data that fall outside the bands? There are multiple data points in each figure panel that are not within the confidence interval bands. Or are these the individuals with large residuals (>2 standard residuals) as per L231-233?

Supplementary material: update manuscript title to the new manuscript.

---

## [Author Response · Author response to Decision Letter 1]

6 Aug 2022

Dear Dr Cristescu

Thank you for taking over from the previous academic editor at such short notice and for the constructive comments on our manuscript entitled ‘Diet similarity among jaguars (Panthera onca) in a high-density population’.

We detail our responses to your comments below.

L158: insert "is" after "this" - Done

L637: "sample" instead of "samples" - Done

L315-317: did you discuss that this result might be a reflection of smaller sample size for females than for males? – We have edited the sentence to clarify this. 

Question: Did the scats confirmed as jaguar originate from adults, subadults, kittens also? The inclusion of scats from young jaguars in the analysis could provide partial explanation of the high taxonomic richness of prey items in the scat of male jaguars. This is something to consider adding to the Discussion – This is an interesting point. Camera trap data from the same trail systems indicate that the trails are monopolised by adult males; this sex bias is reflected in the scats that we found. It seems likely that the scats are produced by the adult males that we detected on camera traps. We have no reason to believe that the high taxonomic richness in the scats is explained by the scats originating from young individuals.

Fig. 1: why not standardize the data by dividing number of prey items by number of scats processed for each jaguar? - Thank you for the suggestion. The goal of Fig. 1 is to illustrate how a few individuals contributed greatly to the dataset, while many individuals contributed minimally to the dataset. As the unit of analysis is “prey item”, we feel that it is more useful to display the number of prey items contributed per individual, rather than the number of prey items standardised by the number of scats collected per individual. 

Fig. 2: add "scats" in brackets - Because scats may contain differing numbers of prey items, the analysis was based on the number of prey items sampled, not the number of scats. Therefore, we have added “(prey items)” to the x-axis in Fig 2, rather than “(scats)”. 

Fig. 3: if you chose to include the jaguar ID for the data that fall outside the confidence interval bands, why not include it for all data that fall outside the bands? There are multiple data points in each figure panel that are not within the confidence interval bands. Or are these the individuals with large residuals (>2 standard residuals) as per L231-233? - Correct, these are the individuals with large residuals (>2 standard residuals). We have included this in the legend for clarity.

Supplementary material: update manuscript title to the new manuscript - Done

We believe that we have fully responded to all the comments. We look forward to hearing from you. 

Yours faithfully

Rebecca Foster, PhD

---

## [Editor Report · Decision Letter 2]

7 Sep 2022

Dietary similarity among jaguars (*Panthera onca*) in a high-density population

PONE-D-21-29787R2

Dear Dr. Foster,

We’re pleased to inform you that your manuscript has been judged scientifically suitable for publication and will be formally accepted for publication once it meets all outstanding technical requirements.

Kind regards,

Bogdan Cristescu

Academic Editor

PLOS ONE

Additional Editor Comments (optional):

The authors incorporated the suggested edits and in one instance provided justification for maintaining the original text.

Congratulations on your paper!

---

## [Editor Report · Acceptance letter]

26 Sep 2022

PONE-D-21-29787R2 

Dietary similarity among jaguars (*Panthera onca*) in a high-density population 

Dear Dr. Foster:

I'm pleased to inform you that your manuscript has been deemed suitable for publication in PLOS ONE. Congratulations! Your manuscript is now with our production department. 

Kind regards, 

on behalf of

Dr. Bogdan Cristescu 

Academic Editor

PLOS ONE